# Pentlandite rocks as sustainable and stable efficient electrocatalysts for hydrogen generation

Bharathi Konkena[1], Kai junge Puring[2], Ilya Sinev[3], Stefan Piontek[2], Oleksiy Khavryuchenko[4], Johannes P. Dürholt[5], Rochus Schmid[5], Harun Tüysüz[6], Martin Muhler[3], Wolfgang Schuhmann[1] & Ulf-Peter Apfel[2]

The need for sustainable catalysts for an efficient hydrogen evolution reaction is of significant interest for modern society. Inspired by comparable structural properties of [FeNi]-hydrogenase, here we present the natural ore pentlandite ($Fe_{4.5}Ni_{4.5}S_8$) as a direct 'rock' electrode material for hydrogen evolution under acidic conditions with an overpotential of 280 mV at 10 mA cm$^{-2}$. Furthermore, it reaches a value as low as 190 mV after 96 h of electrolysis due to surface sulfur depletion, which may change the electronic structure of the catalytically active nickel–iron centres. The 'rock' material shows an unexpected catalytic activity with comparable overpotential and Tafel slope to some well-developed metallic or nanostructured catalysts. Notably, the 'rock' material offers high current densities ($\leq 650$ mA cm$^{-2}$) without any loss in activity for approximately 170 h. The superior hydrogen evolution performance of pentlandites as 'rock' electrode labels this ore as a promising electrocatalyst for future hydrogen-based economy.

[1] Ruhr-Universität Bochum, Analytical Chemistry—Center for Electrochemical Sciences (CES), Universitätsstrasse 150, D-44780 Bochum, Germany.
[2] Ruhr-Universität Bochum, Inorganic Chemistry I, Universitätsstrasse 150, D-44780 Bochum, Germany. [3] Ruhr-Universität Bochum, Industrial Chemistry, Universitätsstrasse 150, D-44780 Bochum, Germany. [4] Slovak University of Technology in Bratislava, Faculty of Materials Science and Technology, Bottova 25, 91724 Trnava, Slovakia. [5] Ruhr-Universität Bochum, Inorganic Chemistry II, Universitätsstrasse 150, D-44780 Bochum, Germany. [6] Max-Planck-Institut für Kohlenforschung, Kaiser-Wilhelm-Platz 1, D-45470 Mülheim an der Ruhr, Germany. Correspondence and requests for materials should be addressed to W.S. (email: wolfgang.schuhmann@rub.de) or to U.-P.A. (email: ulf.apfel@rub.de).

The increasing global need of energy marks the finding of alternative and storable energy sources an important enterprise. In this view, hydrogen is of particular interest since it combines the possibility to efficiently 'store' energy, possesses the highest-energy density of common fuels and reveals a sustainable combustion process[1]. Currently, platinum and its alloys play a dominant role in the $H_2$ evolution reaction (HER) and allow production of $H_2$ at low overpotentials with fast reaction rates and high current densities in acidic electrolytes[2]. The actual price and low natural abundance, however, render platinum a dinosaur in a future sustainable 'hydrogen economy'. Among the numerous suggested non-noble metal HER catalysts, nanostructured transition metal dichalcogenides are promising candidates for this reaction[3–6]. The establishment of $MS_2$-based HER electrocatalysts, however, with satisfactory stability and activity yet remains elusive. Likewise, nanostructured transition metal chalcogenides, for example, $NiSe_2$ nanofibres[7], $MoS_2/CoSe_2$ hybrid catalysts[8], $Ni_3S_2$ nanosheets[9] and $FeS_2$ nanostructures[10], induced a high HER activity in such materials. An improved performance has been observed when these materials were anchored to graphene[5,11–15], Ni/NiO (ref. 16) or gold surfaces[17]. This observation also reveals a significant problem of non-noble metals—their intrinsically low electrical conductivity compared with noble metal HER electrocatalysts. Recently, ultrathin metallic FeNi sulfide nanosheets with unknown composition showed significantly improved HER activity[18]. This material was operated as a HER electrocatalyst with a low overpotential (117 mV) at current densities of $\sim 10$ mA cm$^{-2}$ for 200 h without any loss in activity. Although this material is very effective, the need for specific surface shapes and the concomitant synthetic procedures required render those materials non-economical.

Contrary to such systems, nature utilizes highly efficient machineries to generate $H_2$. Enzymes such as the [FeNi]- as well as the [FeFe]-hydrogenase comprising Ni- and Fe-sulfide containing active sites are well known for their capability to reduce protons to $H_2$ (ref. 19). Notably, electrons are provided to the bimetallic active sites by closely neighbored [4Fe4S] clusters. The property to combine a highly conductive backbone with a well-defined bimetallic catalytic center rather than well-defined nanoparticles, therefore, seems to be a key towards a sustainable and endurable HER catalyst. A material that combines all of those properties is the ore pentlandite, which has the composition $Ni_{4.5}Fe_{4.5}S_8$. Pentlandites show high electronic conductivity and possess Fe- and Ni-centres bridged by sulfur (Fig. 1a)[20,21]. The special assembly of iron, nickel and sulfur, comprising short intermetallic distances (2.51 Å) closely resembles structural

features of both the active site of the [FeNi]-hydrogenase ($d_{NiFe} = 2.573$ Å)[22] and [FeFe]-hydrogenase ($d_{FeFe} = 2.566$ Å)[23]. A striking advantage is that these minerals are a main source for the production of nickel[24]. Thus, they can be hauled from natural deposits at low cost and used without major processing if sufficiently pure.

Herein, we report on the application of pentlandite as electrocatalyst for the HER. We show that this material can be used as 'rock'-electrodes without the need of further surface modifications and they provide high activity and stability at low overpotential for the generation of hydrogen. With the help of density functional theory calculations, we show a plausible pathway for the formation of $H_2$ on the catalyst surface.

## Results

**Synthesis and characterization.** The natural pentlandite ore contains considerable amounts of silicates in the structure. Thus, for better comparison we synthesized $Ni_{4.5}Fe_{4.5}S_8$ from the elements by means of solid-state synthesis at 1,100 °C for 10 h (ref. 24). Powder X-ray diffraction and X-ray photoemission spectroscopic (XPS) measurements were performed to analyse the quality and composition of both materials. Whereas the natural pentlandite showed several reflections besides the main pentlandite phase, the synthetic $Ni_{4.5}Fe_{4.5}S_8$ revealed high purity[20] without the commonly observed monosulfide solid solution phases (Fig. 2a)[25]. Likewise, Mössbauer spectroscopy confirmed the presence of similar iron sites in both materials (Fig. 2b). The synthetic material reveals two distinct different iron sites with isomeric shifts of 0.13 ($\pm 0.02$) and 0.50 ($\pm 0.02$) mm s$^{-1}$ as well as quadrupole couplings of 0.12 ($\pm 0.02$) and 0.13 ($\pm 0.02$), respectively. This observation is in good agreement with literature reports and structural findings showing two different iron sites[26].

The chemical composition of the natural ore is obviously different from the synthetic material. Energy dispersive X-ray spectroscopy (EDX) analysis confirmed the different overall surface composition of both materials (Supplementary Figs 1 and 2). Whereas synthetic pentlandites showed an ideal iron to nickel to sulfur ratio of 1:1:2.25, the natural mineral revealed only negligible small amounts of nickel and contained a significant amount of silicates (Supplementary Figs 1 and 3). This phase is also visible in scanning electron micrographs (SEMs) and is absent in synthetic $Ni_{4.5}Fe_{4.5}S_8$ (Fig. 2c,d). Notably, no appreciable periodical surface features were observed in either case. XPS spectra show that both Fe and Ni in the surface appear severely oxidized in pristine samples (Supplementary Figs 4 and 5;

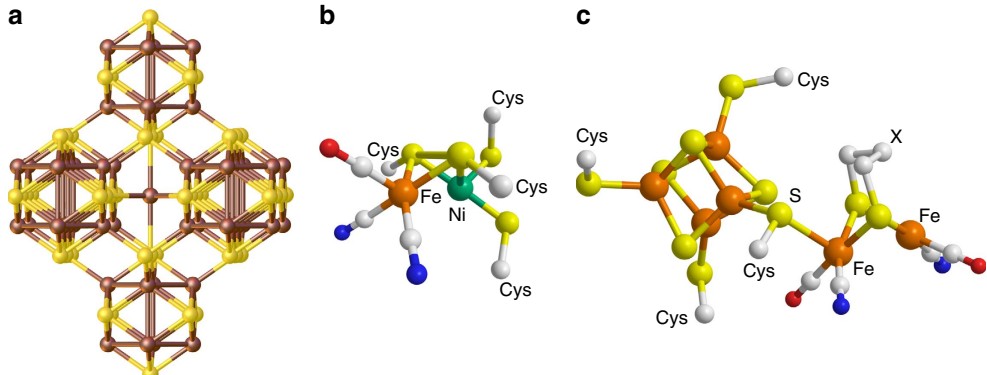

**Figure 1 | Structural comparison of penlandite with hydrogenases.** (**a**) Crystal structure of $Ni_{4.5}Fe_{4.5}S_8$. The nickel and iron sites (brown) share the same positions within the crystal and are bridged by sulfur (yellow). (**b**) Active site of the [FeNi]-hydrogenase (PDB: 4U9H) as well as (**c**) [FeFe]-hydrogenase (X = NH, PDB: 1HFE).

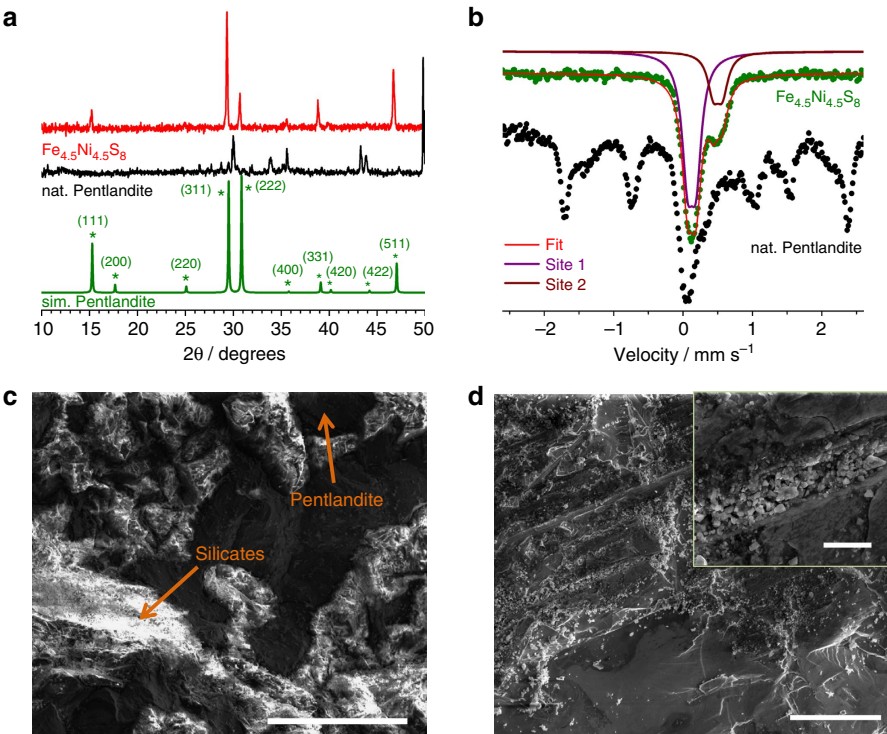

**Figure 2 | Characterization of the pentlandites.** (**a**) Powder X-ray diffraction pattern for natural and synthesized pentlandites. (**b**) Mössbauer spectra of natural and synthesized pentlandites. (**c**) SEM image of the natural pentlandite ore (scale bar, 300 µm). Silicates were assigned by EDX spot analysis. (**d**) SEM image of the as synthesized pentlandite with a composition of $Ni_{4.5}Fe_{4.5}S_8$ (scale bar, 100 µm). Inset shows the image at higher magnification (scale bar, 10 µm).

Supplementary Note 1), reflecting the presence of metal oxide species. Peaks at 706.5 and 852.7 eV can be attributed to metallic states that are in good agreement with values reported for $NiS_2$ (ref. 27) and $FeS_2$ (ref. 28) compounds. The results display the oxidation states of iron and nickel in the synthetic material to be 0 as well as III with the relative ratios provided in Supplementary Table 1. The metallic character of pentlandite as well as the close structural resemblance with the hydrogenases' active sites inspired us to apply the pentlandite 'rock' material as an electrode material for the HER reaction without further surface modification.

**Electrochemical hydrogen evolution.** Electrodes for HER were subsequently prepared from the bulk materials by directly connecting the solid materials with a copper wire surrounded by a Teflon tube using epoxide glue (Supplementary Fig. 6).

Electrochemical HER measurements of as-prepared 'rock electrodes' were performed using a three-electrode cell containing 0.5 M sulfuric acid as electrolyte. Ag/AgCl (3 M KCl) and a platinum wire were utilized as reference and counter electrode, respectively. Figure 3a shows the linear sweep voltammograms obtained at a scan rate of $5\,mV\,s^{-1}$ in argon-saturated solution.

Initial experiments on the natural pentlandite ore exhibited HER activity at an overpotential of $\sim 500\,mV$ and low current density ($\leq 10\,mA\,cm^{-2}$). This poor HER activity of natural pentlandite 'rock' is most likely attributed to the low conductivity of the material due to the presence of large amounts of Si- and Mg-containing phases incorporated into the ore. In contrast to natural pentlandites, synthetic pentlandite 'rock' electrodes with a defined $Ni_{4.5}Fe_{4.5}S_8$ composition revealed a substantially improved electrocatalytic activity at low overpotentials of $\sim 280\,mV$ at a current density of $10\,mA\,cm^{-2}$. In fact, compared with $NiS_2$, $FeS_2$ and $MoS_2$ nanomaterials applied as reference

materials within this study, synthetic pentlandite 'rock' electrodes reveal a superior performance as evidenced from the voltammograms (Fig. 3a; Supplementary Table 2). The essentially higher HER activity of the synthetic pentlandite may be attributed to a larger number of exposed active sites. We hence determined the electrochemical surface area (ECSA) of $NiS_2$, $FeS_2$, $MoS_2$, the natural and the synthetic pentlandite from the electrochemical double-layer capacitance ($C_{dl}$) of the catalysts using cyclic voltammetry (Fig. 3b; Supplementary Fig. 7). The linear slope of the capacitive current as a function of scan rate (Fig. 3c) is equivalent to twice of the double-layer capacitance and represented as ECSA[18]. Notably, the ECSA of the synthetic pentlandite is significantly larger than that of the other materials tested and suggests an increased HER activity of the synthetic pentlandite with a large number of exposed surface sites (Supplementary Fig. 8).

**The influence of surface structure.** We subsequently focused on the synthetic pentlandites as an idealized mineral. Whereas effective non-noble metal-based HER catalysts require nanostructured particles mounted on a conductive electrode[9,18] pentlandites do not require any artificial nanostructuring or grafting to achieve high HER activities at low overpotentials. In contrast to bulk materials, the high surface area of a nanoscaled material is considered to have an impact on the interaction with the surrounding electrolyte and the observed improved catalytic activity. Therefore, understanding the interactions occurring at electrode/electrolyte interfaces at the atomic level is considered to be of utmost importance for the design of effective electrocatalysts. The mere fact that the herein reported pentlandites reveal such high HER activity at low potentials indicates that besides a high surface area commonly arising from nanostructuring additional factors need to be addressed. Notably,

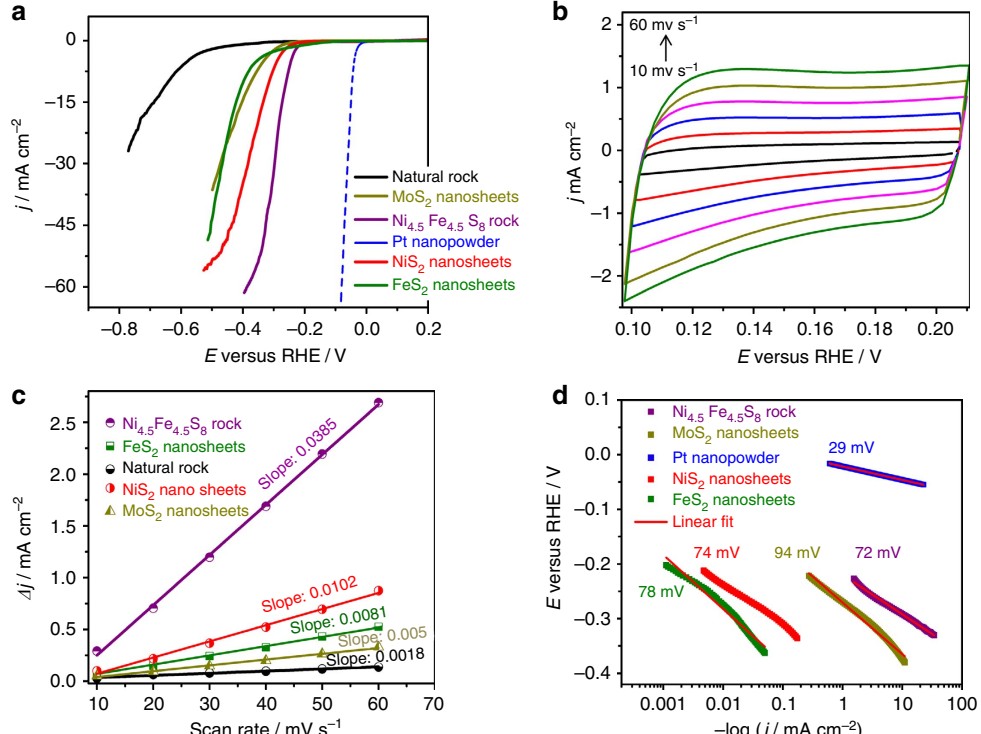

**Figure 3 | Electrochemical characterization. (a)** Linear sweep voltammograms of the catalysts recorded at a sweep rate of $5\,mV\,s^{-1}$ in 0.5 M $H_2SO_4$. The diffusion limited current density was normalized to the geometric area of the electrodes. **(b)** Cyclic voltammograms for the synthetic pentlandite and **(c)** charging current density differences ($\Delta j = j_a - j_c$) as a function of the scan rate. The linear slope is equivalent to twice of the double-layer capacitance $C_{dl}$, representing the electrochemical surface area (ECSA). **(d)** Tafel plots derived from voltammograms at a sweep rate of $1\,mV\,s^{-1}$.

neither measurement with exfoliated nor ball-milled nanosized pentlandite samples revealed any improved electrocatalytic behaviour (Supplementary Figs 9 and 10).

**HER reaction kinetics.** Elementary reaction steps on the electrode surface are a key factor to understand the materials' performance. To understand such steps involved in the HER kinetics, we analysed the Tafel plots ($\eta = \rho \log j + \log j_0$, overpotential $\eta$, current density $j$, Tafel slope $\rho$ and exchange current density $j_0$) that allow for an initial mechanistic hypothesis (Fig. 3d). In general, HER mechanisms can involve the following three steps[29]

$$\text{Volmer}: \quad H_3O^+ + e^- \rightarrow H_{ads} + H_2O \tag{1}$$

$$\left(\sigma = 120\,mV\,dec^{-1}\right)$$

$$\text{Heyrovsky}: \quad H_{ads} + H^+ + e^- \rightarrow H_2 \tag{2}$$

$$\left(\sigma = 40\,mV\,dec^{-1}\right)$$

$$\text{Tafel}: \quad H_{ads} + H_{ads} \rightarrow H_2 \quad \left(\sigma = 30\,mV\,dec^{-1}\right) \tag{3}$$

For Pt nanopowder the Tafel slope is ~29 mV dec$^{-1}$, which is close to the theoretical value ($b = 2.3$ RT/2F) and can be assigned to a HER mechanism including Volmer and Tafel steps. The recombination step (equation (3)) was shown to be the rate-limiting step at low overpotentials, where the chemisorption of hydrogen by the metal is not favoured. For $MoS_2$ nanosheets, a Tafel slope of 95 mV dec$^{-1}$ gives rise to a Volmer- and Heyrovsky-based mechanism with electrochemical desorption of hydrogen as the rate-limiting step. In contrast to $MoS_2$ nanosheets, the synthetic $Ni_{4.5}Fe_{4.5}S_8$ 'rocks' exhibit a Tafel slope of 72 mV dec$^{-1}$. Tafel slopes of 60–70 mV dec$^{-1}$ can be assigned to a fast Volmer-type discharge reaction (equation (1)), followed

by a rate-limiting recombination step (equation (3)), where the chemisorption of hydrogen from aqueous solutions at the electrode surface only requires small activation energy. In the rate-determining process, the adsorbed H atoms migrate over the electrode surface to interact with other adsorbed H atoms to form molecular hydrogen[29].

**Impedance measurements.** We assumed the resistance of the electrocatalyst to be responsible for the surprisingly high activity of the synthetic pentlandite. We therefore performed electro-chemical impedance spectroscopy to characterize the interfacial electrode kinetics and resistance of the material. Figure 4a shows the Nyquist plots of natural (black curve) and $Ni_{4.5}Fe_{4.5}S_8$ pentlandite rocks (red curve) at an applied overpotential of 300 mV versus reversible hydrogen electrode (RHE). The impedance data were fitted to an equivalent circuit (Fig. 4a), employing a constant phase element to determine the charge-transfer resistance $R_{ct}$. The electrochemical impedance spectroscopy data reveal a significantly lower charge-transfer resistance ($R_{ct}$, 57.2 Ω) of the $Ni_{4.5}Fe_{4.5}S_8$ bulk electrode as compared with the natural pentlandite (1.05 kΩ), $MoS_2$ nanosheets (412.4 Ω), $NiS_2$ (213.2 Ω) as well as $FeS_2$ (374.6 Ω) nanoparticles (Supplementary Fig. 11), revealing a faster electron transfer and a higher Faradaic efficiency during HER. We believe that the high HER activity stems from synergetic effects between the bi-transition metals and sulfur sites resulting in a high conductivity as well as the 'right' surface assembly of the catalytic sites. This is consistent with XPS data from sputtered pentlandite, revealing a higher metallic character on 'cleaning' of the electrode surface.

Although $Ni_{4.5}Fe_{4.5}S_8$ reveals an excellent performance as a noble metal-free electrocatalyst, long-time durability under different conditions at high current densities is an important

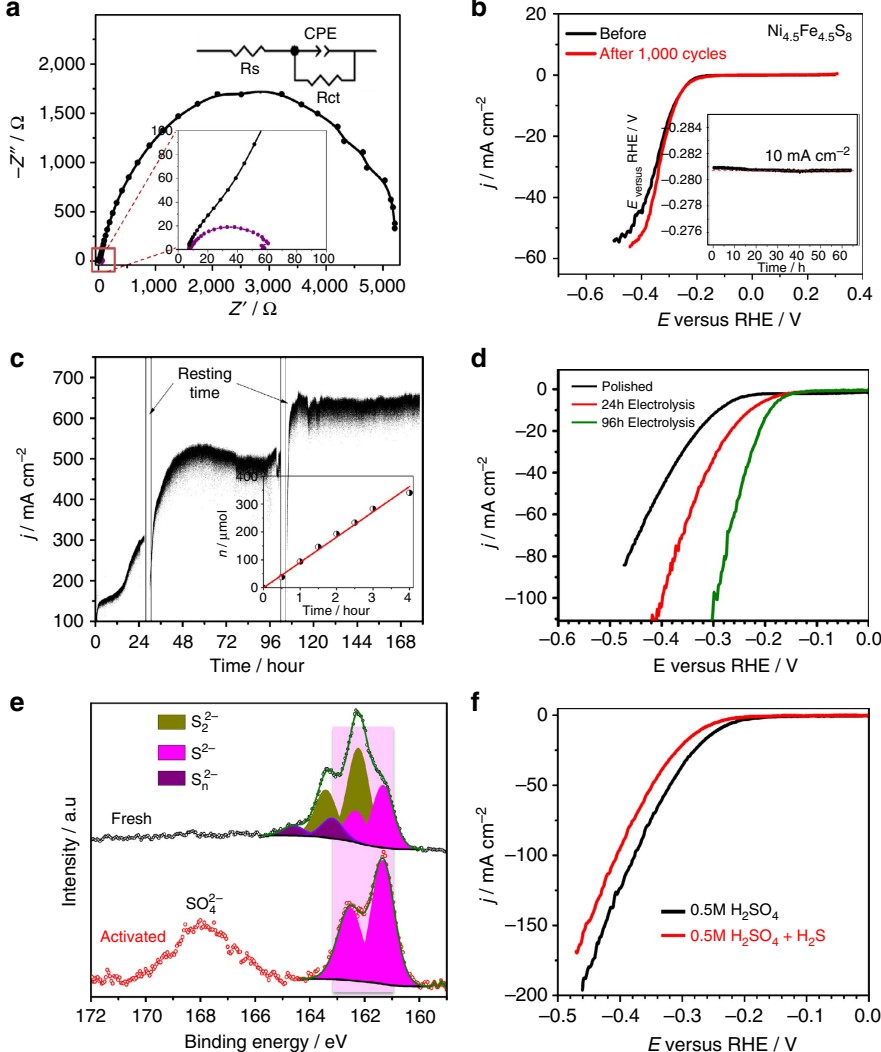

**Figure 4 | Impedance spectroscopy and durability testing.** (**a**) Nyquist plots and the equivalent circuit at HER overpotentials ($\eta = 300$ mV) in 0.5 M $H_2SO_4$. (**b**) HER polarization curves before and after 1,000 voltammetric cycles. The inset shows the potential–time plots at a current density of 10 mA cm$^{-2}$. (**c**) Long-term current-time plot during 170 h electrolysis at an overpotential of 0.6 V. The inset shows a comparison of the amount of measured and calculated $H_2$. (**d**) LSV curves at 0 (black), 24 (red) and 96 h (green) electrolysis. (**e**) S 2p core-level spectra of a 'rock' electrode before and after 24 h of electrolysis. (**f**) Inactivation of a pentlandite electrode upon addition of $H_2S$.

demand for its potential application. Hence, the long-term stability of the $Ni_{4.5}Fe_{4.5}S_8$ electrode was investigated by electrochemical cycling in 0.5 M $H_2SO_4$ in a potential range between $-0.4$ and $0.4$ V versus RHE for 1,000 cycles at 50 mV s$^{-1}$. No appreciable change in the catalytic performance after 1,000 cycles was observed indicating excellent electrochemical stability of the $Ni_{4.5}Fe_{4.5}S_8$ electrode (Fig. 4b). In addition, constant-potential electrolysis supported this finding and revealed HER at a current density of 10 mA cm$^{-2}$ for at least 60 h in 0.5 M $H_2SO_4$ as shown in the inset of Fig. 4b.

**Durability and activation studies**. To determine the Faradaic efficiency and to study the long-term stability at higher current densities, we performed HER experiments at $Ni_{4.5}Fe_{4.5}S_8$ 'rocks' for 170 h (Fig. 4c) at a constant overpotential of 0.6 V. The increase in the $H_2$ gas concentration was quantitatively monitored by gas chromatography with a thermal conductivity detector (GC-TCD) during the first 4 h giving rise to 2.14 mmol h$^{-1}$ cm$^{-2}$ of $H_2$ (inset of Fig. 4c) produced during electrolysis at $Ni_{4.5}Fe_{4.5}S_8$ 'rocks'. This amount of $H_2$ is surprising and comparable to that of Pt-based electrolysers that were shown to afford

11 mmol h$^{-1}$ cm$^{-2}$ at a constant potential of $-1.2$ V (versus RHE) with a Faradaic efficiency of 94% (ref. 30). Similar values for non-noble metal HER catalysts were never observed for any 'rock'-like bulk compound before. Whereas chronoamperometric experiments at a current density of 10 mA cm$^{-2}$ revealed a constant potential over time, the performed electrolysis at 0.6 V overpotential versus RHE showed an unexpected phenomenon (Fig. 4c). Notably, with time, the current density increases and a constant current was only observed after $\sim 48$ h. This indicates that the electrode surface can be activated under reductive conditions at more negative potentials leading concomitantly to higher HER activity. This activation could, in principle, further lead to a downshift of the onset potential for the HER reaction and is evidenced from the linear sweep voltammograms (Fig. 4d; Supplementary Fig. 12). Subsequently, $H_2$ formation occurs with overpotentials of only $\sim 190$ mV at 10 mA cm$^2$ current density. A likely explanation for this phenomenon is the depletion of surface sulfur from the material. However, Pt oxidation and subsequent deposition at the working electrode was reported to show a similar behaviour[31]. To clarify this hypothesis, we performed analogous electrochemical experiments applying

a glassy carbon counter electrode (Supplementary Note 2). Similar to our observations with the Pt counter electrode, a decrease of the overpotential for HER (Supplementary Fig. 13) as well as an increase of activity was observed. This observation clearly confirms that the activation is not a result from a deposition of Pt on the working electrode and more likely stems from depletion of surface sulfur from the catalyst, thus, affording more exposed nickel–iron surface sites[21]. During the activation process, the surface structure and overall composition of the electrode remained unaltered (Supplementary Figs 14 and 15). We therefore investigated S $2p$ core-level XPS spectra of the electrode before and after electrolysis (Fig. 4e; Supplementary Fig. 16). Contrary to pentlandite electrodes before electrolysis, the activated sample solely showed sulfur $2p$ doublets at the lower binding energies of 161.3 and 162.5 eV attributable to the $S^{2-}$ ion (Fig. 4e) and the amount of disulfide-type sulfur ($S_2^{2-}$) was significantly decreased resulting in sulfur vacancies on the catalyst surface[32]. Such a behaviour inevitably promotes the reactivity of the exposed Ni–Fe sites and thus the HER performance. Along this line, treatment with additional sulfides thus can be expected to reverse the activation of the electrode. We hence investigated the activated electrodes in $H_2S$ saturated 0.5 M $H_2SO_4$ solutions for HER (Fig. 4f). The onset potential shifts by 30 mV to more negative potentials suggesting occupation of S vacancies and further supporting the reversible activation by S depletion.

**Hydrogen quantification and alternative proton sources.** In addition, the Faradaic efficiency ($\eta$) of the $Ni_{4.5}Fe_{4.5}S_8$ 'rock' electrode was determined for the first 4 h to be 91 ± 5% confirming the high activity of the electrode material to efficiently catalyse the HER reaction in 0.5 M $H_2SO_4$. Next, we investigated the influence of the pH value on the turnover frequency from the slope of the Clark electrode signal during the first hour of the catalytic HER at different pH values and normalized to the amount of $H_2$ quantified by gas chromatography measurements (Supplementary Fig. 17). Notably, the high activity is not restricted to sulfuric acid solutions. Different tests with strong acids such as HCl, $HNO_3$, HBr, as well as $H_3PO_4$ revealed $H_2$ evolution at comparable overpotentials and current densities. Likewise, acetic acid as well as citric acid were tested as potential proton sources and allowed for moderate $H_2$ generation, which is expected due to their lower $pK_a$ value and the subsequent lower ionic strength (Supplementary Fig. 18).

Commonly noble metal catalysts are severely poisoned by sulfur-containing species present in waste acidic materials[33]. We hence simulated sulfur-poisoning conditions by performing proton reduction in $H_2S$ atmosphere. The presence of $H_2S$ did not affect the catalytic $H_2$ evolution notably. We noticed that although revealing a smaller activity, $H_2S$ can be directly utilized as a proton source for $H_2$ evolution. It is only limited by its low

solubility ($\sim 0.2$ M in $H_2O$) and acidity ($pK_a = 7$; Supplementary Fig. 19).

**Theoretical investigations.** To gain some insight into the roles of Ni, Fe and S in promoting the HER activity theoretical investigations on $Ni_{4.5}Fe_{4.5}S_8$ were performed. Since full-scale modelling of a $Ni_{4.5}Fe_{4.5}S_8$ surface is computationally extremely time demanding, a zeroth-order model was considered in a cluster approach with main structural features extracted from the bulk and saturated by protons and water molecules.

The crystal structure of pentlandites can be best described as a $M_8S_6$ cuboctahedra interconnected by tetrahedrally bound sulfur and additional in-plane metal atoms (Supplementary Note 3). The exposure of the $M_8S_6$ cuboctahedra surface results in significant structural changes of this unit (Supplementary Fig. 20), large distortions of the metal framework with a motif that is highly flexible and sensitive to protonation as well as change of charge/spin state (Supplementary Table 3). The relevant structures during the $H_2$ formation are shown in Fig. 5. For the potential hydrogen formation, optimizations of systems protonated at different water molecules bound to the metals were performed (Supplementary Fig. 21). In all cases, the proton was transferred from the hydronium ion to a metal atom leading to a hydride between a nickel and an iron atom (Fig. 5a). For the case with the largest negative charge on the hydrogen, a reaction path for the hydrogen formation was determined by a series of constrained geometry optimizations, using the shortest H–H distance as a reaction coordinate (Supplementary Fig. 22). During this process, another proton that was originally bonded to a sulfur atom is pulled towards the hydride (Fig. 5b) and leads to the formation of $H_2$ as an exothermic process ($\Delta E = -6.4\,kcal\,mol^{-1}$; Fig. 5c). Sulfur vacancies created during electrolysis may change the electronic structure of the catalytically active Ni–Fe centres and thus facilitate the HER process. These results for the model system corroborate the potential of pentlandite to electrocatalyse the HER process with a mechanism strikingly similar to the biological [FeNi]-hydrogenase[34].

## Discussion

We have successfully synthesized pentlandites with a composition of $Fe_{4.5}Ni_{4.5}S_8$, exhibiting superior activity and stability for the HER. Notably, whereas exhaustive nanoparticle and electrode preparation is commonly required, the high conductivity of $Fe_{4.5}Ni_{4.5}S_8$ allows for a direct application as 'rock' electrode without any additional artificial surface structuring. The prepared non-noble metal electrodes reveal a high endurance in corrosive solutions such as sulfuric acid at an overpotential of 280 mV at 10 mA cm$^{-2}$, but reaches lower values as low as 190 mV after 96 h of electrolysis due to sulfur depletion from the surface creates

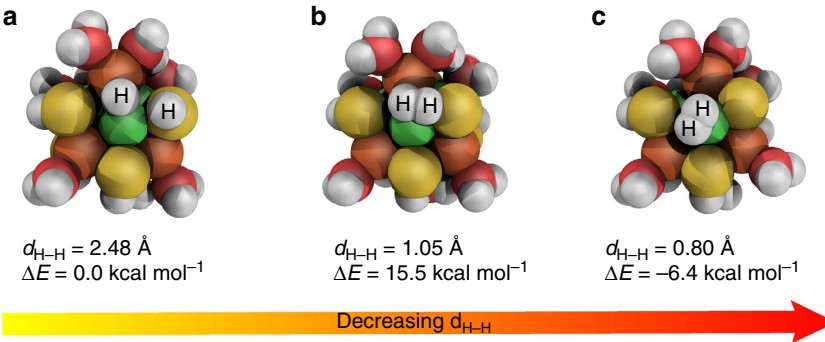

a  $d_{H-H} = 2.48$ Å  $\Delta E = 0.0$ kcal mol$^{-1}$

b  $d_{H-H} = 1.05$ Å  $\Delta E = 15.5$ kcal mol$^{-1}$

c  $d_{H-H} = 0.80$ Å  $\Delta E = -6.4$ kcal mol$^{-1}$

Decreasing $d_{H-H}$

**Figure 5 | HER mechanism.** $H_2$ formation on an exposed $H_7Ni_4Fe_4S_6(H_2O)_8^{3+}$ cluster. (**a**) The cluster with the bridging hydride, (**b**) the first order transition state and (**c**) the cluster with formed $H_2$ (colour scheme: green: nickel; white: hydrogen; red: oxygen; brown: iron; yellow: sulfur).

sulfur vacancies. Most likely this depletion changes the electronic structure of the catalytically active Ni–Fe centres facilitating the HER process. The HER activity of the pentlandite 'rock' electrodes reveal comparable overpotential and Tafel slope in comparison with some of well-developed metallic catalysts. Notably, these electrodes even allow $H_2$ evolution under catalyst poisoning conditions. In addition, the electrodes can be operated at current densities as high as $650\,mA\,cm^{-1}$ for 170 h without loss in activity. Altogether, the natural ore pentlandite is a highly abundant, cheap, robust and highly efficient electrocatalyst for the HER reaction that will allow for a significant boost of non-noble metal catalysts.

## Methods

**Synthesis of pentlandite.** Natural pentlandite from Intsilä, Finland, was obtained from the mineral and 'rock' collection of the Department of Geology, Mineralogy and Geophysics at the Ruhr-University in Bochum. The synthesis of $Fe_{4.5}Ni_{4.5}S_8$ was performed from the elements. The high-purity elements iron (1.67 g, 29.8 mmol), nickel (1.75 g, 29.8 mmol) and sulfur (1.70 g, 53.1 mmol) were ground until a homogeneous mixture was obtained. This mixture was placed in a 10 mm silica tube. Subsequently, the silica tube was sealed under a static vacuum and was heated to 700 °C with $5\,°C\,min^{-1}$. After 3 h of annealing at 700 °C, the temperature was raised to 1,100 °C within 30 min. After 10 h at 1,100 °C, the mixture was allowed to cool down to room temperature.

**Physical characterization.** SEM images and EDX analysis were performed using a Quanta 3D FEG scanning electron microscope (FEI) operated at 20 kV for imaging and at 4.4 kV for EDX analysis. X-ray diffraction measurements were recorded using a Bruker AXS Advance powder diffractometer (40 kV, 50 mA, sealed Cu-$K\alpha$ X-ray tube) equipped with a graphite monochromator. The diffraction pattern was collected in the continuous scan mode at a scan rate of 0.03° per 5 s.

Zero-field Mössbauer spectra were recorded at 298.15 K using a SeeCo constant acceleration spectrometer equipped with a temperature controller maintaining temperatures within ± 0.1 K and a $^{57}Co$ radiation source in a Rh matrix. Isomer shifts are referred to α-Fe metal at room temperature. Data were fit with a sum of Lorentzian quadrupole doublets using a least-squares routine with WMOSS program.

XPS measurements were carried out in an ultrahigh vacuum set-up equipped with a monochromatic Al $K_\alpha$ X-ray source ($hv = 1486.6\,eV$), operated at 14.5 kV and 35 mA, and a high-resolution Gammadata-Scienta SES 2002 analyser. The base pressure in the measurement chamber was maintained at $\sim 5 \times 10^{-10}$ mbar. The measurements were carried out in the fixed transmission mode with a pass energy of 200 eV, resulting in an overall energy resolution of 0.25 eV. A flood gun was applied to compensate the charging effects. High-resolution spectra for C 1s, O 1s, S 2p, and Fe 2p and Ni 2p photoelectron lines were recorded. The C 1s signal of the adventitious carbon was an external standard attributed to 284.5 eV binding energy. The Casa XPS software with a pseudo-Voigt Gaussian-Lorentzian product (oxide species) or asymmetric LA (metallic states) functions and Shirley background subtraction was used for peak deconvolution. Atomic ratios were calculated from XPS intensities corrected to Scofield photoemission cross-sections. Measurements were carried out on pristine synthetic pentlandite and after sputtering for 30 and 60 min. Sputtering was carried out in a preparation chamber with Ar$^+$ accelerated to 2.5 keV using SPECS IQE 11/35 ion source.

**Exfoliation of bulk $Ni_{4.5}Fe_{4.5}S_8$ crystals.** Bulk-layered $Ni_{4.5}Fe_{4.5}S_8$ crystals were exfoliated by dispersing $5\,mg\,ml^{-1}$ of the crystals in cetyltrimethylammonium bromide (CTAB) surfactant solution ($2\,mg\,ml^{-1}$), in water followed by sonication for 10 h in a 100 W tip sonicator. After sonication, the dispersions were subjected to differential centrifugation to narrow down the size distribution. In a typical method, the dispersions were centrifuged at 1,000 r.p.m. for 1 h, and the supernatant was separated and subjected to successive centrifugation at 4,000 and 6,000 r.p.m. for periods of 2 h. The process was terminated at this stage (at 6,000 r.p.m.); the final sediment was collected and redispersed in water under sonication and used for further investigations.

**Ball milling of bulk $Ni_{4.5}Fe_{4.5}S_8$.** The planetary ball mill (Fritsch Pulverisette 7, classic line) with a SiC milling containers (diameter: 2.5 cm, volume: 13 cm$^3$) and four SiC milling balls (diameter: 1 cm) was used to milled the bulk material. In a typical milling process, $\sim 1$ g of bulk $Ni_{4.5}Fe_{4.5}S_8$ was milled for duration of 10 min at a speed rate of 650 r.p.m.

**Synthesis of $MoS_2$ nanosheets.** Bulk $MoS_2$ crystals were synthesized by chemical vapour transport method. In a typical synthesis, elemental powders of Mo and S were mixed in stoichiometric proportions (1:1) and inserted into a quartz tube. The quartz tube was evacuated to $\sim 10^{-6}$ mbar and sealed. The sealed quartz tube was placed in a tube furnace at 800 °C for 2 weeks to ensure the crystal formation. The

quartz tube was cooled down to room temperature and opened for collecting the formed crystals (Supplementary Fig. 23).

**Liquid exfoliation of $MoS_2$ sheets.** Bulk-layered $MoS_2$ crystals were exfoliated by dispersing $5\,mg\,ml^{-1}$ of the crystals in CTAB surfactant solution ($2\,mg\,ml^{-1}$), in water followed by a sonication for 10 h in a 100 W bath sonicator. After sonication, the dispersions were subjected to differential centrifugation to narrow down the size distribution. In a typical method, the dispersions were centrifuged at 1,000 r.p.m. for 1 h, and the supernatant was separated and subjected to successive centrifugation at 2,000 and 4,000 r.p.m. for periods of 2 h. The process was terminated at this stage (at 4,000 r.p.m.); the sediment was collected and redispersed in water under sonication. After sonication the dispersion was stable for 3 months without any flocculation and used for further investigations (Supplementary Fig. 24).

**Synthesis of $NiS_2$ nanoparticles.** $NiS_2$ nanosheets were synthesized by a one-step hydrothermal method. In a typical method, 4 mmol of nickel chloride hexahydrate ($NiCl_2 \cdot 6H_2O$) and 4 mmol of $Na_2S_2O_3 \cdot 5H_2O$ were mixed in a beaker containing 30 ml of milliQ water and stirred for 1 h. The mixed solution was transferred to a 60 ml Teflon-lined stainless steel autoclave and heated for 24 h at 180 °C. The precipitate was collected by centrifugation and repeatedly washed with ethanol and water (1:2) mixture, and then dried (Supplementary Fig. 25).

**Synthesis of $FeS_2$ nanoparticles.** $FeS_2$ nanosheets were synthesized by a step hydrothermal method. In a typical method, 4 mmol of ferric chloride tetrahydrate ($FeCl_2 \cdot 4H_2O$) and 4 mmol of $Na_2S_2O_3 \cdot 5H_2O$ were mixed in a beaker containing 30 ml of milliQ water and stirred for 1 h. The mixed solution was transferred to a 60 ml Teflon-lined stainless steel autoclave and heated for 24 h at 180 °C. The precipitate was collected by centrifugation and repeatedly washed with ethanol and water (1:2) mixture, and then dried (Supplementary Fig. 26).

**Electrode fabrication.** Synthetic or natural pentlandite 'rocks' were cut into blocks of $\sim 0.5 \times 0.5 \times 0.5$ cm and placed in a crimp that was soldered to a copper cable. Subsequently, the assembly was housed in a Teflon tube, insulating the copper wire from solution. The tip of the prepared electrode was then covered with epoxide glue, dried and polished to reveal solely a pure pentlandite surface as potential electrode. The polished electrode was then directly used in the electrochemical measurements (Supplementary Fig. 6). The geometric area of the prepared electrode was 0.135 cm$^2$.

**Electrochemical measurements.** Electrochemical testing of the catalyst was conducted using a standard three-electrode set-up using a GAMRY Reference 600 Potentiostat or an Autolab potentiostat/galvanostat (PGSTAT12). The catalyst materials were directly used in bulk as the working electrode. Ag/AgCl (saturated KCl) or Ag/AgCl (3 M KCl) and a Pt grid were used as reference and counter electrode, respectively. If not otherwise stated, 0.5 M $H_2SO_4$ was used as the electrolyte and the measured potential was converted to the RHE potential according to $E_{RHE} = E_{Ag/AgCl} + X + 0.059$ pH ($X = +0.197\,V$) (saturated KCl) or 0.210 V (3 M KCl).

Before each measurement, the cell was electrochemically washed by cycling at least five times in the potential window $-0.4\,V < E_{Ag/AgCl} < 0.4\,V$ at a scan rate of $100\,mV\,s^{-1}$ until a stable cyclic voltammogram was obtained. Linear sweep voltammetry was then performed at a scan rate of $5\,mV\,s^{-1}$. Long-term durability measurements were done using controlled potential coulometry at $E_{RHE} = -0.6\,V$ monitoring the transferred charge and the current. Chronopotentiometric measurements (galvanostatic electrolysis) were performed at a current density of $10\,mA\,cm^{-2}$ for 60 h. Electrochemical impedance spectroscopy was recorded in the frequency range from 50 kHz to 1 Hz at the corresponding open-circuit potential and HER overpotential of the electrode using an ac perturbation of $10\,mV_{pp}$. The resistance of the solution was determined from the resulting Nyquist plot. All measurements were carried out at room temperature.

Gas samples were directly taken from the headspace of the electrochemical cell after a specific time interval and injected into a Shimadzu GC-2010 gas chromatograph. The data were then evaluated using a calibration curve for the correlation between the peak area and the hydrogen amount.

**Electrochemical measurements of powder or nanosized samples.** All electrochemical measurements were performed using an Autolab potentiostat/galvanostat (PGSTAT12, Eco Chemie) in a conventional three-electrode cell in combination with a speed control unit (CTV101) and a rotating disk electrode rotator (EDI101; Radiometer). A disc-shaped glassy carbon electrode of geometric area 0.126 cm$^2$ modified with the catalysts was used as working electrode, a Ag/AgCl/3 M KCl as reference electrode and a platinum mesh as counter electrode. The reference electrode was calibrated with respect to the RHE. before the experiments, the glassy carbon electrode was polished on a polishing cloth using different alumina pastes (3.0–0.05 μm) to obtain a mirror-like surface followed by ultrasonic cleaning in water. For electrochemical measurements, the catalyst ink was prepared by

dispersing $5.0\,mg\,ml^{-1}$ of the catalyst in water followed by ultra-sonication for 30 min. A volume of $5.0\,\mu l$ of the catalyst suspension was drop coated onto the polished glassy carbon electrode and dried in air at room temperature. Before the HER measurements, modified electrodes were subjected to continuous potential cycling in the potential window of $-0.5$ to $0.5\,V$ versus Ag/AgCl/3 M KCl, until reproducible voltammograms were obtained.

**Determination of the Faradaic efficiency.** The Faradaic efficiency was determined according to equation (4)

$$\eta_{H2}^{eff} = \frac{nF[H_2]}{I \times t} \qquad (4)$$

with $n$ number of transferred electrons, F is Faradaic constant, $[H_2]$ is detected concentration of hydrogen, $I$ is current and $t$ is time.

**Calculations.** All quantum-chemical calculations were performed within the framework of density functional theory, employing the Perdew, Burke and Ernzerhof (PBE0) hybrid functional[35–37] together with Ahlrichs' double-zeta split valence all electron basis sets[38]. Further details on the computational methods are given in the Supplementary Information (Supplementary Note 4).

**Data availability.** The authors declare that the data supporting the findings of this study are available within the article and its Supplementary Information files, and from the authors on reasonable request.

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

## Acknowledgements

We thank Sandra Schmidt for SEM images. Financial support by the Deutsche Forschungsgemeinschaft (DFG) in the framework of the Cluster of Excellence Resolv (EXC1069) is gratefully acknowledged. U.-P.A. is grateful for the financial support by the Fonds of the Chemical Industry (Liebig grant to U.-P.A.) and the Deutsche Forschungsgemeinschaft (Emmy Noether grant to U.-P.A., AP242/2-1).

## Author contributions

B.K., K.j.P., S.P. and U.-P.A. designed and performed the synthesis and all electrochemistry experiments. I.S. and M.M. performed XPS measurements and analysed the data. O.K., J.P.D. and R.S. performed the calculations. H.T. planned and performed ball-milling experiments. B.K. and U.-P.A. wrote the manuscript. W.S. and U.-P.A. are responsible for planning and supervision.

## Additional information

**Competing financial interests:** The authors declare no competing financial interest.

