## [Peer Review File · Nature Communications]

Reviewers' comments:

Reviewer #1 (Remarks to the Author):

In this manuscript, the authors report the excellent electrocatalytic activity for the hydrogen evolution reaction on a synthetic Ni_{0.45}Fe_{0.45}S₈ "rock" electrode. The authors have done a very thorough experimental and theoretical study, and the manuscript is very well written. Given that (1) to the best of my knowledge this is the first report of the hydrogen evolution catalytic activity on Ni_{0.45}Fe_{0.45}S₈, (2) the authors have demonstrated high catalytic activity for Ni_{0.45}Fe_{0.45}S₈, (3) the authors have also demonstrated high stability of their "rock" electrode, and (4) the authors show that they can use this material as a "rock" electrode without nano-structuring; it is my opinion that this manuscript warrants publication in Nature Communications. However, below are a few suggestions for minor revisions to the manuscript.

1. After Figure 3, the authors state "In fact, synthetic pentlandite rock electrodes reveal a superior performance compared to the best known noble-metal free HER catalysts, e.g. MoS₂ nanosheets as evidenced from the voltammograms (Figure 3a)." From Figure 3 in the authors' manuscript, their MoS₂ structure has an over potential of ~0.4V at 10 mA/cm², and they report a Tafel slope of 94 mV/decade. This looks correct for the 2H-MoS₂ structure, however the 1T-MoS₂ structure is known to have Tafel slopes around 40 mV/decade and over potentials of approximately 0.25V at 10 mA/cm² in strong acid (see work by Jin (Wisconsin) and Chhowalla (Rutgers)). Thus, I suggest that the authors comment on whether they are comparing their results to the 2H or 1T structure of MoS₂ and remove the claim that it is superior to the "best known noble-metal free HER catalysts."

2. One of the interesting results that the authors report is that the performance of the Ni_{0.45}Fe_{0.45}S₈ rock electrode increases with increasing electrolysis time. Material characterization measurements (e.g. SEM, EDX, XPS) of the rock electrode after the electrolysis would be very interesting and add value to this section of the manuscript.

3. Figures 4C and D show the effect of pH on the electrocatalytic performance. It is my opinion that this section be moved into the Supporting Information. pH comparisons are troublesome due to buffer conditions, changes in ionic strength, and separating proton reduction from water reduction, and thus I feel this section would be more appropriate in Supporting Information.

Reviewer #2 (Remarks to the Author):

In this manuscript, the authors studied the natural and synthetic pentlandite rocks (Fe_{4.5}Ni_{4.5}S₈) on HER and found they were promising electrocatalyst for hydrogen based economy. The work is interesting. The catalyst shows good durability even though the activity is not so good. However, the authors should address the following many questions after major revisions before it can be reconsidered for publication:

1. Why the current density increased dramatically during the chronoamperometry testing? The authors think it is due to the depletion of oxides or surficial sulfides from the catalyst surface which generates more metallic nickel or iron on the catalyst surface. To support this opinion, the authors compared the XPS data of synthesized pentlandite before and after Ar⁺ sputtering. This Reviewer thinks comparing the XPS data of pentlandite before and after the chronoamperometry testing would be more direct and convictive. Moreover, to exclude another possible reason that the generation of nanostructure on the catalyst surface which lead to the increase of activity after 96h electrolysis, the morphology, crystal structure and chemical structure changes of the pentlandite surface is necessary.

2. More careful interpretation of such high durability is important for better presenting the current

catalyst.

3. Does the $\text{Fe}_{4.5}\text{Ni}_{4.5}\text{S}_8$ pentlandite rock show better performance than the nanostructure $\text{Fe}_{4.5}\text{Ni}_{4.5}\text{S}_8$? If not, why we should synthesize the $\text{Fe}_{4.5}\text{Ni}_{4.5}\text{S}_8$ pentlandite rock for the HER since the synthesis procedure of such $\text{Fe}_{4.5}\text{Ni}_{4.5}\text{S}_8$ pentlandite rock is not simpler or easier than synthesizing nanostructure materials.

4. The authors said "the $\text{Ni}_{4.5}\text{Fe}_{4.5}\text{S}_8$ rock exhibits a Tafel slope of 72 mV/dec. Thus, a fast Volmer-type discharge reaction (Eq. 1) followed by a rate-limiting recombination step (Eq. 3) similar to Pt electrodes is assumed". As we know and the authors described in the Eq.1-3, the Tafel slopes for Eq2 (Heyrovsky) is about 40 mV/dec, while that for Eq.3 (Tafel) is 30 mV/dec. Herein, the $\text{Ni}_{4.5}\text{Fe}_{4.5}\text{S}_8$ rock exhibits a Tafel slope of 72 mV/dec, which is closer to 40 mV/dec rather than 30 mV/dec. Why the authors propose the rate limiting step is the recombination step (Eq.3)?

5. The authors observed that "the overpotential of 280 mV at 10 mA/cm² was reduced to 190 mV after 96 h of electrolysis". And the authors gave an explanation to be "the depletion of oxides or surficial sulfides from the catalyst surface that affording more metallic nickel-iron exposed surface sites" So, 1). Do the authors mean that the catalytic active/ or more active sites are metallic nickel-iron rather than the Fe(III)-Ni(II)? However, in natural FeNi-/FeFe- hydrogenase, the metal sulfides are the catalytic active sites; 2). Do the authors think that the formed nickel-iron exposed surface sites are in nanoscale that improved the performance?

6. In experiments, the $\text{Ni}_{4.5}\text{Fe}_{4.5}\text{S}_8$ are used, while for theoretical calculations, M8S6 model was used instead. Why did the authors make this change?

7. In general, more explanations about why the authors choose $\text{Ni}_{4.5}\text{Fe}_{4.5}\text{S}_8$ as an efficient HER material should be provided. In other words, there is a lack of explanation about how the authors know this particular $\text{Ni}_{4.5}\text{Fe}_{4.5}\text{S}_8$ is an efficient HER material.

8. In the text, "...pentlandites are stable in a broad pH range and exhibit high temperature stability..." but in the following text, no experimental details or previous literature have been provided, so the authors need to do something to prove it.

9. In Fig. 2c and 2d, the SEM images the authors provided seem too hard to identify the Silicates and Pentlandite part, so the TEM text is necessary. There are also some mistakes such as the two figure 2c onn page 7: "Aside of the silicate phase, both pentlandite phases reveal similar surface properties (Figure 2c and 2c)".

10. There are several mistakes in serial number of figures. Such as "Figure 1b", "Figure S1 and S2" and "Figure 2c and 2c", these serial numbers don't match the Figure.

11. In the text "...a high surface area arising from nano-structuring additional factors need to be addressed... did not reveal any improved electrocatalytic behavior." Firstly, powders do not means nano size, so how the authors use Figure S5 and Figure S6 to prove nano structuring did not reveal any improvement of electrocatalytic behavior. Secondly, "the high surface area of a nano-scaled material is considered to have an impact on the interaction with the surrounding electrolyte and the observed improved catalytic activity." So why the nano structuring do not show a better performance in $\text{Ni}_{4.5}\text{Fe}_{4.5}\text{S}_8$? The reasons should be presented.

12. The authors put "no need for nano " in the title to express that the natural material is good enough for HER. But in the SEM images there are still some nanostructures in the natural and manmade materials. These nanostructures may be important in catalysis and thuc contradict the statement of "no need for nano". In fact, the performance is not as good as some nanomaterials reported in the literature.

13. In Figure 4, the authors mentioned that after 96 hours the performance of the catalyst

increased, which need some explanations. Maybe the increase came from the nanostructure formed on the surface, who knows?

Reviewer #3 (Remarks to the Author):

The manuscript reported the exploration of pentlandite as the electrocatalyst for hydrogen production. They provide electrochemical results and calculations to support their claims of excellent electrocatalytic activity for HER. However, similar NiFe hydrogenases are researched as the HER catalysts, more importantly, except the report of this pentlandite mine catalyst, the current electrocatalyst didn't present impressive activity for HER compared to other reported HER electrocatalysts (J. Am. Chem. Soc. 2015, 137, 11900). Thus, the limited novelty of this work prevents the acceptance of this work in Nature Commun (J. Am. Chem. Soc. 2005, 127, 14871). Moreover, the calculation part should be enhanced to understanding hydrogen adsorption at the active sites. Especially, the relationship of structure/surface and activity/stability of NiFeS_x should be analyzed in the revised manuscript. Based on the limited significance and novelty of this work, rejection of this work is suggested, after considering the concerns, the manuscript could be submitted to a more specialized journal related to materials.

1, in the part of the surface structure analysis, this phase is also visible in scanning electron micrographs (SEM) and is absent in synthetic Ni_{4.5}Fe_{4.5}S₈. Aside of the silicate phase, both pentlandite phases reveal similar surface properties (Figure 2c and 2c)." it is hard to find the surface difference from the SEM images for readers. Moreover, more characterization of NiFeS_x should be provided to analyze their activity mechanism.

2, the authors compared the synthetic pentlandite rock electrodes Ni_{4.5}Fe_{4.5}S₈ to the HER catalysts of MoS₂ nanosheets and claimed the high HER activity of Ni_{4.5}Fe_{4.5}S₈ stemmed from synergetic effects between the bi-transition metals and sulfur sites. It is more reasonable to compare Ni_{4.5}Fe_{4.5}S₈ to Ni-S or Fe-S catalysts but not MoS₂.

3, The synthetic pentlandite Ni_{4.5}Fe_{4.5}S₈ was active in bulk form by comparing with exfoliated pentlandite samples, maybe nanostructured Ni_{4.5}Fe_{4.5}S₈ should be prepared and compared with the bulk one.

4. to analyze the HER kinetics, the authors said "a fast Volmer-type discharge reaction (Eq. 1) followed by a rate-limiting recombination step (Eq. 3) similar to Pt electrodes is assumed" this HER kinetics (by analysis the Tafel slope) is quite different with that of the Pt, the claim of similar mechanism with Pt needs second thought.

5, the authors didn't present convincing evidence and explanation about enhanced activity during the cycling test. For the electrochemical measurement, Pt is not suitable to use as the counter electrode (J. Mater. Chem. A, 2015,3, 13080). The electrochemical test details should be introduced. The activity of comparison with other reported HER catalyst should be provided.

Reviewer #1

In this manuscript, the authors report the excellent electrocatalytic activity for the hydrogen evolution reaction on a synthetic Ni_{0.45}Fe_{0.45}S₈ "rock" electrode. The authors have done a very thorough experimental and theoretical study, and the manuscript is very well written. Given that (1) to the best of my knowledge this is the first report of the hydrogen evolution catalytic activity on Ni_{0.45}Fe_{0.45}S₈, (2) the authors have demonstrated high catalytic activity for Ni_{0.45}Fe_{0.45}S₈, (3) the authors have also demonstrated high stability of their "rock" electrode, and (4) the authors show that they can use this material as a "rock" electrode without nano-structuring; it is my opinion that this manuscript warrants publication in Nature Communications. However, below are a few suggestions for minor revisions to the manuscript.

1. After Figure 3, the authors state "In fact, synthetic pentlandite rock electrodes reveal a superior performance compared to the best known noble-metal free HER catalysts, e.g. MoS₂ nanosheets as evidenced from the voltammograms (Figure 3a)." From Figure 3 in the authors' manuscript, their MoS₂ structure has an over potential of ~0.4V at 10 mA/cm², and they report a Tafel slope of 94 mV/decade. This looks correct for the 2H-MoS₂ structure, however the 1T-MoS₂ structure is known to have Tafel slopes around 40 mV/decade and over potentials of approximately 0.25V at 10 mA/cm² in strong acid (see work by Jin Wisconsin and Chhowalla (Rutgers)). Thus, I suggest that the authors comment on whether they are comparing their results to the 2H or 1T structure of MoS₂ and remove the claim that it is superior to the "best known noble-metal free HER catalysts."

The HER catalytic activity of MoS₂ nanosheets reported in the present work corresponds to the 2H-MoS₂ structure. The 2H-MoS₂ structure is evident from the PXRD pattern (JCPDF # 37-1492). The PXRD spectra of 2H-MoS₂ are now provided as a supplementary Figure S23. The points suggested by the reviewer have been corrected in the revised manuscript.

2. One of the interesting results that the authors report is that the performance of the Ni_{0.45}Fe_{0.45}S₈ rock electrode increases with increasing electrolysis time. Material characterization measurements (e.g. SEM, EDX and XPS) of the rock electrode after the electrolysis would be very interesting and add value to this section of the manuscript.

We repeated our electrolysis experiments and characterized the 'rock-electrode' by SEM, EDX and XPS after 24 hours to back up our hypothesis with solid data. We clearly show that the sulfur content on the electrode surface decreases over time resulting in sulfur vacancies and giving deeper insight on the observed improved HER activity of synthetic $Ni_{0.45}Fe_{0.45}S_8$ rock electrode with time. The results obtained after the electrolysis are now part of the revised manuscript and supportive figures are shown in the supplementary part.

- 3. Figures 4C and D show the effect of pH on the electrocatalytic performance. It is my opinion that this section be moved into the Supporting Information. pH comparisons are troublesome due to buffer conditions, changes in ionic strength, and separating proton reduction from water reduction, and thus I feel this section would be more appropriate in Supporting Information.**

We do share the objections of the reviewer and placed the data relevant to the effect of pH on the HER electrocatalytic performance in the supporting information.

Reviewer #2

In this manuscript, the authors studied the natural and synthetic pentlandite rocks ($Fe_{4.5}Ni_{4.5}S_8$) on HER and found they were promising electrocatalyst for hydrogen based economy. The work is interesting. The catalyst shows good durability even though the activity is not so good. However, the authors should address the following many questions after major revisions before it can be reconsidered for publication:

- 1. Why the current density increased dramatically during the chronoamperometry testing? The authors think it is due to the depletion of oxides or surficial sulfides from the catalyst surface which generates more metallic nickel or iron on the catalyst surface. To support this opinion, the authors compared the XPS data of synthesized pentlandite before and after Ar^+ sputtering. This Reviewer thinks comparing the XPS data of pentlandite before and after the chronoamperometry testing would be more direct and convictive. Moreover, to exclude another possible reason that the generation of nanostructure on the catalyst surface which lead to the increase of activity after 96h electrolysis, the morphology, crystal structure and chemical structure changes of the pentlandite surface is necessary.**

A careful XPS, SEM and EDX investigation on the rock electrode after the electrolysis for 24 hours was added. These results support our assumption of sulfur depletion.

2. More careful interpretation of such high durability is important for better presenting the current catalyst.

We extended and more carefully interpreted our results now provided in the revised manuscript.

3. Does the Fe_{4.5}Ni_{4.5}S₈ pentlandite rock show better performance than the nanostructure Fe_{4.5}Ni_{4.5}S₈?

To check this point we attempted the synthesis of a nano-phase of the Fe_{4.5}Ni_{4.5}S₈ pentlandite by ball milling of bulk pentlandite. The SEM and TEM data of the obtained product are provided in the SI part (Figure S9). We furthermore added electrochemical data on the HER activity of pentlandite nanostructured material that does not show any difference compared to the bulk material.

If not, why we should synthesize the Fe_{4.5}Ni_{4.5}S₈ pentlandite rock for the HER since the synthesis procedure of such Fe_{4.5}Ni_{4.5}S₈ pentlandite rock is not simpler or easier than synthesizing nanostructure materials.

One of the most important limitations of nanostructured catalysts especially in long-term applications is the tight binding to an electrode support. Thus, a huge advantage of a rock-type electrode is seen in the fact that no addition of binder materials is needed for electrode preparation. Actually, as is clearly stated in the manuscript no electrode preparation at all (e.g. drop coating etc.) is required. With this in mind and knowing that pentlandites are usually available in nature and are used for the production of Ni it is obvious that finding electrocatalysts with natural consistency is advantageous.

4. The authors said "the Ni_{4.5}Fe_{4.5}S₈ rock exhibits a Tafel slope of 72 mV/dec. Thus, a fast Volmer-type discharge reaction (Eq. 1) followed by a rate-limiting recombination step (Eq. 3) similar to Pt electrodes is assumed". As we know and the authors described in the Eq.1-3, the Tafel slopes for Eq2 (Heyrovsky) is about 40 mV/dec, while that for Eq.3 (Tafel) is 30 mV/dec. Herein, the Ni_{4.5}Fe_{4.5}S₈ rock exhibits a Tafel slope of 72 mV/dec, which is closer to 40 mV/dec rather than 30 mV/dec. Why the authors propose the rate limiting step is the recombination step (Eq.3)?

For the benefit of the reader, and to avoid any misunderstanding, we have presented the complete mechanism for HER on a metal surface under acidic conditions which is summarized below:

For metals electrolytic hydrogen evolution reaction gives a Tafel line of low slope; $b = 2.3RT/2F$, i.e., 29 mV/decade at 25°C. This value is experimentally well established on platinum and corresponds to a fast discharge step Volmer reaction (Eq 1) followed by a rate limiting combination reaction (Eq. 3). This Tafel slope assumes that the surface coverage with adsorbed H atoms is less (< 0.1) where the chemisorption of H by the metal is not activated. Tafel slopes between 30 to 50 does not require activation energy for the chemisorption of hydrogen from the aqueous solutions by the metals.

For Tafel slope, $b = 2.3RT/2\beta F$ accounted for the moderate surface coverage of H atoms the heat of adsorption and activation energies for adsorption and desorption of H are taken into consideration. Where β is the ratio of variation of the activation energy of desorption of hydrogen with the coverage to the variation of the heat of adsorption with coverage. Giving values for the slopes are summarized in the table. The probable value for β , representing a symmetrical energy barrier for the combination reaction. If the β value is 0.5, the tafel slope is 58 mV/decade, i.e., activated adsorption of H has been found which is corresponding to the appearance of a moderate activation energy for H adsorption due to increasing surface coverage with H atoms. The Tafel slope of 73 mV represents only a small activation energy required for the adsorption of H atoms from aqueous solution with a very low surface coverage with H atoms would correspond to a less active species giving activated hydrogen adsorption.

	Tafel slope (mV/decade)	Examples
$\beta = 1$	29	Pt, Pd
$\beta = 0.68$	43	Pd
$\beta = 0.52$	55	Rh
$\beta = 0.5$	58	Fe

$\beta = 0.45$	65	Ag, Fe, Ni, W
$\beta = 0.4$	73	Au, Fe, Cu, Ag, Mo
$\beta = 0.2$	90	Mo, W

For Tafel slopes usually associated between 60 to 70, a combination mechanism is rate determining and the surface is approaching saturation. Then the combination step may become first order, in which the rate controlling step reaction (ii) is the migration of H atoms on the surface to meet either another immobile adsorbed H atom or an adsorbed H atom which is also mobile on the electrode surface to give a molecular hydrogen.

In contrast to Pt, for metals with higher Tafel slopes (i.e., 60 to 70mV/dec) combination reaction (iii) is the rate determining step.

The discussion relevant to the Tafel analysis was corrected in the revised manuscript.

5. The authors observed that "the overpotential of 280 mV at 10 mA/cm² was reduced to 190 mV after 96 h of electrolysis". And the authors gave an explanation to be "the depletion of oxides or surficial sulfides from the catalyst surface that affording more metallic nickel-iron exposed surface sites" So, 1). Do the authors mean that the catalytic active/ or more active sites are metallic nickel-iron rather than the Fe(III)-Ni(II)? However, in natural FeNi-/FeFe- hydrogenase, the metal sulfides are the catalytic active sites;

Neither did we suggest that pure metallic iron or nickel nor Fe(III)-Ni(II) is responsible for the higher activity. We solely suggest that the removal of additional surface sulfur opens possible reaction sites that do allow for the higher reactivity observed. This is in line with the FeNi and FeFe enzymes. E.g. the [FeFe] hydrogenase from HydA1 possesses an open coordination site at the [2Fe2S] cluster that is responsible for the effective reduction of protons to hydrogen. The redox states that are responsible for the reduction of protons are in both cases low-valence. Please refer to U.-P. Apfel et al., [FeFe] Hydrogenase Models:

an Overview in Bioinspired Catalysis 2014, Wiley-VCH or F. A. Armstrong et al., Chem. Rev. 2007, 107, 4366 for a more detailed mechanism on hydrogenases. A direct comparison of both mechanisms is however misleading and highly speculative and was thus not added in the manuscript.

5.2. Do the authors think that the formed nickel-iron exposed surface sites are in nanoscale that improved the performance?

We think that this effect is on a molecular scale and most certainly spreads over the entire surface over time. Partially occurring nano-scaled environments are therefore not the major reason for the improved activity. This statement is supported by our SEM data that do not show any nano-sized clusters on the surface. Please also refer to Figure S13.

6. In experiments, the Ni_{4.5}Fe_{4.5}S₈ are used, while for theoretical calculations, M₈S₆ model was used instead. Why did the authors make this change?

Since no periodic DFT calculations could be performed due to high computational costs and the difficulties to handle charged systems, a cluster had to be cut out of the bulk material. Note that the additional metal atom, leading to the sum formula Ni_{4.5}Fe_{4.5}S₈ (or M₉S₈) in the bulk material is due to an additional metal atom connecting the M₈S₆ cluster units (see Fig. 1). As explained in the manuscript, the “saturated” M₈S₆ cluster is employed in these model calculations as the zeroth-order approximation of the reactive surface site.

7. In general, more explanations about why the authors choose Ni_{4.5}Fe_{4.5}S₈ as an efficient HER material should be provided. In other words, there is a lack of explanation about how the authors know this particular Ni_{4.5}Fe_{4.5}S₈ is an efficient HER material.

By adding more data to this manuscript we do believe this point became clearer.

8. In the text, "...pentlandites are stable in a broad pH range and exhibit high temperature stability..." but in the following text, no experimental details or previous literature have been provided, so the authors need to do something to prove it.

The sentence was deleted since it was based on our own observations.

9. In Fig. 2c and 2d, the SEM images the authors provided seem too hard to identify the Silicates and Pentlandite part, so the TEM text is necessary. There are also some mistakes such as the two figure 2c on page 7: "Aside of the silicate phase, both pentlandite phases reveal similar surface properties (Figure 2c and 2c)".

To identify the silicate phase in the natural pentlandite we measured XRD and SEM/EDX spectra and these results were already previously shown in the main manuscript. To further support these results we measured the XPS spectra of the natural pentlandite sample and the XPS survey spectra is now provided in the revised manuscript (supplementary part Figure S3), which indeed provides corroborative evidence for the existence of silicates in the natural pentlandite. Due to the thickness of our materials ("rock") a TEM is not feasible and will not give any significant information.

The numbering mistake suggested by the reviewer has been corrected in the revised manuscript.

10. There are several mistakes in serial number of figures. Such as "Figure 1b", "Figure S1 and S2" and "Figure 2c and 2c", these serial numbers don't match the Figure.

The mistakes were corrected in the revised manuscript.

11. In the text "...a high surface area arising from nano-structuring additional factors need to be addressed...did not reveal any improved electrocatalytic behavior." Firstly, powders do not means nano size, so how the authors use Figure S5 and Figure S6 to prove nano structuring did not reveal any improvement of electrocatalytic behavior. Secondly, "the high surface area of a nano-scaled material is considered to have an impact on the interaction with the surrounding electrolyte and the observed improved catalytic activity." So why the nano structuring do not show a better performance in Ni_{4.5}Fe_{4.5}S₈? The reasons should be presented.

We are in complete agreement with the reviewer's point of view concerning the HER activity of the exfoliated powder sample. We now added further data on nano-scaled pentlandite to support our statement. In addition, the reasons for this different behavior are clearly presented within the manuscript. We assume that no nano-structuring is required since the herein reported material is highly conductive and reveals the right surface composition. As such one might say that this material has properties like a simple platinum wire, which is already as a polished bulk material very reactive. Contrary to

pentlandites or platinum, bulk FeS₂, NiS₂, MoS₂, etc. possess a high intrinsic resistance preventing an effective electron transfer.

12. The authors put "no need for nano " in the title to express that the natural material is good enough for HER. But in the SEM images there are still some nanostructures in the natural and manmade materials. These nanostructures may be important in catalysis and thuc contradict the statement of "no need for nano". In fact, the performance is not as good as some nanomaterials reported in the literature.

We are puzzled about this statement. It seems that the referee wants to see a large scale of nanoparticles where there are none. The "nanostructures" he suggests to be in the SEM are still of μm -size. In addition we do provide clear evidence that nanoparticles do not provide an altered activity compared to the bulk material.

The referee should please indicate the material he refers to! A detailed list of catalysts is added to the supplementary part. We are surprised by his statement on performance – usually electrocatalysts are not fully characterized (see table), do not work as bulk material and require a difficult nano-structuring. As we show – this manipulations are not in the pentlandite case necessary and still we obtain a superior electrocatalysts.

13. In Figure 4, the authors mentioned that after 96 hours the performance of the catalyst increased, which need some explanations. Maybe the increase came from the nanostructure formed on the surface, who knows?

A detailed explanation to this point is now added in the main manuscript. The activity stems unequivocally from sulfur vacancies and is not due to nano-structuring.

Reviewer #3

The manuscript reported the exploration of pentlandite as the electrocatalyst for hydrogen production. They provide electrochemical results and calculations to support their claims of excellent electrocatalytic activity for HER. However, similar NiFe hydrogenases are researched as the HER catalysts, more importantly, except the report of this pentlandite mine catalyst, the current electrocatalyst didn't present impressive activity for HER compared to other reported HER electrocatalysts (J. Am. Chem. Soc. 2015, 137, 11900). Thus, the limited novelty of this work prevents the acceptance of this work in Nature Commun (J. Am. Chem. Soc. 2005, 127, 14871). Moreover, the calculation part should be enhanced to understanding hydrogen adsorption at the active

sites. Especially, the relationship of structure/surface and activity/stability of NiFeS_x should be analyzed in the revised manuscript. Based on the limited significance and novelty of this work, rejection of this work is suggested, after considering the concerns, the manuscript could be submitted to a more specialized journal related to materials.

We did not report on the catalytic activity of NiFe hydrogenases as HER catalysts and we referred to NiFe hydrogenases only in terms of a reasoning, why pentlandite could be assumed to have catalytic activity for HER. Our aim was not to show the most active HER catalyst ever reported, but to demonstrate that a rock-type material which is not exfoliated into a nanostructured material exhibits surprisingly high catalytic activity together with a very impressive catalytic stability. As the referee is for sure aware, catalytic activity as expressed by minimum overpotential for a give direction is always compromised by stability as especially seen for biological systems. Taking this into consideration, we feel that the first application of pentlandite rocks as solid electrodes together with the demonstrated characteristics including the stability is of high novelty in the field. The first manuscript mentioned by the authors is dealing with “Metallic Iron–Nickel Sulfide Ultrathin Nanosheets As a Highly Active Electrocatalyst for Hydrogen Evolution Reaction in Acidic Media” which evidently differs from our proposed approach by using nanosheets, and the second mentioned manuscripts entitled “Catalysts for Hydrogen Evolution from the [NiFe] Hydrogenase to the Ni₂P(001) Surface: The Importance of Ensemble Effect” is a theoretical study of a number of HER catalysts, however only slightly related to the proposed pentlandite rock electrodes. Having said this, we do not see how the mentioned manuscripts are lowering the novelty of the proposed approach. The last sentence of the referee “Especially, the relationship of structure/surface and activity/stability of NiFeS_x should be analyzed in the revised manuscript” is evidently addressing the holy grail of electrocatalysis. As the referee is for sure aware, the surface structure of an electrocatalysts is modulated at the applied potential and hence the surface structure of the working catalyst cannot be elucidated with the presently available techniques. As pointed out before, the relationship between activity and stability is usually antagonistic and the novelty of the proposed catalyst lies in the fact that is provided a rock-type electrode surface with high activity and simultaneously high stability.

- 1. In the part of the surface structure analysis, this phase is also visible in scanning electron micrographs (SEM) and is absent in synthetic Ni_{4.5}Fe_{4.5}S₈. Aside of the silicate phase, both pentlandite phases reveal similar surface properties (Figure 2c and 2c)." it is hard to find the surface difference from the SEM images for**

readers. Moreover, more characterization of NiFeS_x should be provided to analyze their activity mechanism.

To identify the silicate phase in the natural pentlandite we measured XRD and SEM/EDX spectra and these results were previously provided in the main manuscript. To further support these results we now measured the XPS spectra of the natural pentlandite sample and XPS survey spectra (Figure S3) is now provided in the revised manuscript which indeed provide corroborative evidence for the existence of silicates in the natural pentlandite whereas synthetic pentlandite shows a pure-phase.

- 1. The authors compared the synthetic pentlandite rock electrodes Ni_{4.5}Fe_{4.5}S₈ to the HER catalysts of MoS₂ nanosheets and claimed the high HER activity of Ni_{4.5}Fe_{4.5}S₈ stemmed from synergetic effects between the bi-transition metals and sulfur sites. It is more reasonable to compare Ni_{4.5}Fe_{4.5}S₈ to Ni-S or Fe-S catalysts but not MoS₂.**

To receive some perspective about the performance of our catalyst compared to related materials reported in the literature, we endeavored to synthesize NiS₂ and FeS₂ nanomaterials and evaluated their electrocatalytic activity for HER under similar conditions to Ni_{4.5}Fe_{4.5}S₈. The methodology used to synthesize NiS₂ and FeS₂, and corresponding characterization details were provided in the supplementary data and a comparison of the electrochemical parameters is provided in the main manuscript.

- 2. The synthetic pentlandite Ni_{4.5}Fe_{4.5}S₈ was active in bulk form by comparing with exfoliated pentlandite samples, maybe nanostructured Ni_{4.5}Fe_{4.5}S₈ should be prepared and compared with the bulk one.**

Results are provided within the supporting information.

- 3. To analyze the HER kinetics, the authors said "a fast Volmer-type discharge reaction (Eq. 1) followed by a rate-limiting recombination step (Eq. 3) similar to Pt electrodes is assumed" this HER kinetics (by analysis the Tafel slope) is quite different with that of the Pt, the claim of similar mechanism with Pt needs second thought.**

See comments to referee 2 (Question Number 4).

- 4. The authors didn't present convincing evidence and explanation about enhanced activity during the cycling test. For the electrochemical measurement, Pt is not suitable to use as the counter electrode (J. Mater. Chem. A, 2015, 3, 13080). The electrochemical test details should be introduced.**

We re-performed our experiments using a glassy carbon electrode and placed the data into the main manuscript as well as supporting information. It is obvious that a similar activity enhancement can be observed. In addition, we backed up our data by additional SEM, XPS and EDX measurements, clearly showing no Pt deposited on the electrode.

- 5. The activity of comparison with other reported HER catalyst should be provided.**
A detailed list of other HER catalysts and their performance reported in literature was added and is provided as supplementary Table S2.

REVIEWERS' COMMENTS:

Reviewer #1 (Remarks to the Author):

In the revised version of the manuscript, the authors sufficiently addressed my concerns, and it appears that they also addressed most of the concerns of the other referees. However, one additional comment that arose from the new manuscript is that in lines 164-167 the authors state, "Notably, the ECSA of the synthetic pentlandite is significantly larger than that of the other materials tested and suggests an increased HER activity of the synthetic pentlandite with a large number of exposed surface sites." This is interesting because the structure which is not "nano" is the one that has the highest electrochemical surface area. I suggest to normalize the surface area effect, the authors include (in supporting information) a figure analogous to Figure 3a, where the current density is normalized to ECSA of the catalyst. This would provide additional information on the catalytic performance.

Reviewer #2 (Remarks to the Author):

The authors have improved the manuscript, but for Nature Communications, it is still not up to the standard as to novelty, understanding and rigor. Also the notion of "no nano" in title is misleading other than simply catching the eyes since it is not proved and it can not be general, ...

Instead, other specialized journal may serve a better purpose for the work.

Reviewer #3 (Remarks to the Author):

The authors have tried to address the questions/concerns raised by the reviewers and revised their manuscript substantially. I think it is publishable.

I do feel that this material, as a bulk rock material, exhibit activity towards HER; by increasing surface area or surface roughness (go to nano), the activity should be improved as the activity is reported based on geometric area and will surely improve with the increased surface area (or roughness). So I would suggest the authors tune down their claim on this aspect.

REVIEWERS' COMMENTS:

Reviewer #1 (Remarks to the Author):

In the revised version of the manuscript, the authors sufficiently addressed my concerns, and it appears that they also addressed most of the concerns of the other referees. However, one additional comment that arose from the new manuscript is that in lines 164-167 the authors state, "Notably, the ECSA of the synthetic pentlandite is significantly larger than that of the other materials tested and suggests an increased HER activity of the synthetic pentlandite with a large number of exposed surface sites." This is interesting because the structure which is not "nano" is the one that has the highest electrochemical surface area. I suggest to normalize the surface area effect, the authors include (in supporting information) a figure analogous to Figure 3a, where the current density is normalized to ECSA of the catalyst. This would provide additional information on the catalytic performance.

We thank the reviewer for his supportive feedback. Such a figure was added to the supporting part as was requested by the reviewer.

Reviewer #2 (Remarks to the Author):

The authors have improved the manuscript, but for Nature Communications, it is still not up to the standard as to novelty, understanding and rigor.

The reviewer states that our manuscript does not meet the standard as to novelty, understanding and rigor for publication in Nature Communications, however, without providing any scientific reason nor any other reason. We disagree with this view, which is also opposed to the other two reviewers.

Also the notion of "no nano" in title is misleading other than simply catching the eyes since it is not proved and it can not be general, ...

We removed the phrase "no need for nano" from the title.

Instead, other specialized journal may serve a better purpose for the work.

We believe that our manuscript is suitable for publication and we are supported in our view by the other two reviewers.

Reviewer #3 (Remarks to the Author):

The authors have tried to address the questions/concerns raised by the reviewers and revised their manuscript substantially. I think it is publishable.

We thank the reviewer for his supportive statement.

I do feel that this material, as a bulk rock material, exhibit activity towards HER; by increasing surface area or surface roughness (go to nano), the activity should be improved as the activity is reported based on geometric area and will surely improve with the increased surface area(or roughness). So I would suggest the authors tune down their claim on this aspect.

We agree with this objections. As stated in the response to reviewer two, we removed the statement “no need for nano” from the title. In addition to this removal, we tuned the phrases within the paper, so that it reads that ‘no artificial processing’ is required. We hope that by highlighting, that we aim at showing that the material can be used and activated without artificial and difficult nano-particle preparation.